# Defining Metaniches in the Oral Cavity According to Their Microbial Composition and Cytokine Profile

**DOI:** 10.3390/ijms21218218

**Published:** 2020-11-03

**Authors:** Corinna L. Seidel, Roman G. Gerlach, Patrick Wiedemann, Matthias Weider, Gabriele Rodrian, Michael Hader, Benjamin Frey, Udo S. Gaipl, Aline Bozec, Fabian Cieplik, Christian Kirschneck, Christian Bogdan, Lina Gölz

**Affiliations:** 1Department of Orthodontics and Orofacial Orthopedics, Universitätsklinikum Erlangen and Friedrich-Alexander Universität (FAU) Erlangen-Nürnberg, Glückstr. 11, 91054 Erlangen, Germany; patrick.wiedemann@fau.de (P.W.); matthias.weider@uk-erlangen.de (M.W.); gabriele.rodrian@uk-erlangen.de (G.R.); lina.goelz@uk-erlangen.de (L.G.); 2Mikrobiologisches Institut-Klinische Mikrobiologie, Immunologie und Hygiene, Universitätsklinikum Erlangen and Friedrich-Alexander-Universität (FAU) Erlangen-Nürnberg, Wasserturmstraße 3/5, 91054 Erlangen, Germany; roman.gerlach@uk-erlangen.de (R.G.G.); christian.bogdan@uk-erlangen.de (C.B.); 3Department of Radiation Oncology, Universitätsklinikum Erlangen and Friedrich-Alexander-Universität (FAU) Erlangen-Nürnberg, Universitätsstr. 27, 91054 Erlangen, Germany; michael.hader@uk-erlangen.de (M.H.); benjamin.frey@uk-erlangen.de (B.F.); udo.gaipl@uk-erlangen.de (U.S.G.); 4Medical Immunology Campus Erlangen, FAU Erlangen-Nürnberg, 91054 Erlangen, Germany; 5Department of Medicine 3, Rheumatology and Immunology, Universitätsklinikum Erlangen and Friedrich-Alexander-Universität (FAU) Erlangen-Nürnberg, Glückstr. 6, 91054 Erlangen, Germany; aline.bozec@uk-erlangen.de; 6Department of Conservative Dentistry and Periodontology, University Hospital Regensburg, Franz-Josef-Strauß-Allee 11, 93053 Regensburg, Germany; fabian.cieplik@ukr.de; 7Department of Orthodontics, University Hospital Regensburg, Franz-Josef-Strauß-Allee 11, 93053 Regensburg, Germany; christian.kirschneck@ukr.de

**Keywords:** oral microbiota, oral niches, metaniches, cytokine profile, oral biofilms and fluids, next-generation sequencing, multiplex immunoassay

## Abstract

The human oral microbiota consists of over 700 widespread taxa colonizing the oral cavity in several anatomically diverse oral niches. Lately, sequencing of the 16S rRNA genes has become an acknowledged, culture-independent method to characterize the oral microbiota. However, only a small amount of data are available concerning microbial differences between oral niches in periodontal health and disease. In the context of periodontitis, the cytokine expression in the gingival crevicular fluid has been studied in detail, whereas little is known about the cytokine profile in hard and soft tissue biofilms. In order to characterize oral niches in periodontal health, the oral microbiota and cytokine pattern were analyzed at seven different sites (plaque (P), gingival crevicular fluid (GCF), saliva (S), tongue (T), hard palate (HP), cheek (C) and sublingual area (U)) of 20 young adults using next-generation sequencing and multiplex immunoassays. Site-specific microbial compositions were detected, which clustered into three distinct metaniches (“P-GCF”, “S-T-HP” and “C-U”) and were associated with niche-/metaniche-specific cytokine profiles. Our findings allow the definition of distinct metaniches according to their microbial composition, partly reflected by their cytokine profile, and provide new insights into microenvironmental similarities between anatomical diverse oral niches.

## 1. Introduction

The oral cavity serves as the first part of the digestive system and the respiratory tract showing complex environmental conditions and anatomically diverse oral niches, such as supra- and subgingival plaque on teeth, gingival sulcus and gingival crevicular fluid (GCF), keratinized gingiva and buccal mucosa, tongue, sublingual area, hard/soft palate, lips, tonsils and saliva [1].

In 2015, Hajishengallis et al. [2] introduced a new explanatory model in the etiology of periodontal disease: periodontal health was defined as a state of “symbiosis” between the resident, eubiotic microorganisms and the immune cells of the host. Instead, periodontitis was defined as a “dysbiotic inflammatory disease”, in which exogenous factors (e.g., plaque accumulation, smoking, malnourishment, stress) can lead to an imbalance between pathogenic and protective bacteria causing dysregulation of the immune response, an excessive inflammatory reaction and the development of periodontal disease [2]. Therefore, the investigation of both the oral microbiota and the cytokine profile is crucial when studying periodontal health and periodontitis.

The oral cavity shows the second-largest microbiota in the human body after the gut microbiota [1,3,4]. Recently, sequencing of the 16S rRNA genes has become an acknowledged, culture-independent method to characterize the oral microbiota, as most of the oral bacteria cannot be cultivated [1]. 775 taxa have been detected up to now, while only 57% are cultivable and named, 30% are uncultivable phylotypes, and 13% are cultivable but unnamed [5,6]. As the intrauterine surrounding is sterile, the development of the neonatal microbiota begins during and after birth, which is an ongoing process during childhood towards an “adult oral microbiota”. This process seems to be influenced by the cross-talk between the oral and gut microbiota as well as by exogenous factors, such as the microbiota of family members or the type of diet. The oral-gut microbiota axis also seems to be crucial for the maturation of our immune system, especially during the first year of life [3,7,8,9,10,11,12]. Regarding bacterial composition in patients with periodontitis and periodontal health, most studies have focused on the characterization of the whole microbiota or investigation of supra- or subgingival plaque [1,13], while only a few elucidated microbial similarities or dissimilarities between distinct oral niches with varying—and at times contradictory—results [4,14,15,16,17,18]. Moreover, there is a lack of a clear definition and differentiation of “symbiosis” and “dysbiosis”, complicating the usage of this definition [2] for diagnosis and research purposes.

The immune response against bacterial pathogens in the oral cavity is triggered by several immune cells and humoral mediators like cytokines and chemokines [19]. Inflammation in periodontal disease has been studied intensively by analysis of the cytokine expression in GCF [20,21]. Thereby an increase of pro-inflammatory cytokines such as interleukin (IL)-1β, -2, -6, -8, tumour-necrosis-factor (TNF), and interferon-gamma (IFN-γ) was seen in periodontitis, while a decrease of inflammation was detected after periodontal treatment [19,20,21,22,23,24]. Moreover, several studies focused on salivary cytokine biomarkers and detected higher levels of pro-inflammatory cytokines not only in periodontitis [25] but also in the saliva of smokers [26] and patients with oral squamous cell carcinoma [27]. However, to our knowledge, so far no correlation analysis of cytokine and chemokine expression with differences in the oral microbiome in periodontal health has been performed. Furthermore, the complex cytokine network and their pathways in the activation of immune cells in oral diseases are crucial [28] and further studies are necessary to gain knowledge about cytokine composition in other oral niches like dental biofilm or the salivary pellicle on different soft tissue surfaces.

Therefore, this study aimed to analyse the microbial composition and the cytokine expression in different oral niches of young adults maintaining periodontal health and excellent oral hygiene (Approximal-Plaque-Index <25%, Bleeding-on-Probing-Index <3%, study population consisting of dental students and dentists with an excellent daily oral hygiene and regular prophylaxis) and to investigate possible similarities or discrepancies in the oral cavity. Defining oral sites in perfect periodontal health is crucial for future studies analysing changes in the composition of the oral microbiota or immune response associated with treatment or oral diseases.

## 2. Results

Oral microbiota composition and cytokine profiles were studied in 280 samples collected from 7 different oral niches (2 samples per niche) of 20 study participants. To this aim, unstimulated saliva samples (S) were obtained by the spitting method and soft tissue samples were collected from the middle part of the hard palate (HP), the middle and anterior part of the tongue (T), the middle part of the sublingual area (U) and the molar region of the right cheek (C) using sterile swabs. Moreover, hard tissue samples of supragingival plaque (P) were taken by sterile curettes and samples of gingival crevicular fluid (GCF) were collected from 6 periodontal pockets using sterile paper strips.

### 2.1. Microbial Composition

Oral microbiota analyses of 140 samples from 7 different oral niches (P, GCF, S, T, HP, C, U) showed 9 different microbial phyla, 14 classes, 30 orders, 44 families, and 75 genera derived from 300 operational taxonomic units (OTU).

In summary, the oral microbiota of healthy young adults is dominated by *Actinobacteriota, Bacteroidota, Firmicutes* and *Proteobacteria* on phylum level, *Actinobacteria, Bacilli, Bacteroidia, Fusobacteria* and *Gammaproteobacteria* on class level, *Actinomycetales, Bacteroidales, Flavobacteriales, Fusobacteriales, Lactobacillales, Pasteurellales* and *Veillonellales* on order level; *Actinomycetaceae, Camplyobacteriae, Carnobacteriacae, Flavobacteriacae, Fusobacteriaceae, Gemellaceae, Lachnospraceae, Leptotrichiaceae, Micrococcaceae, Porphyromonadacea, Prevotellaceae, Streptococcaceae* and *Veillonellaceae* on family level, and *Actinobacillus, Actinomyces, Alloprevotella, Campylobacter, Capnocytophaga, Corynebacterium, Fusobacterium, Gemella, Granulicatella, Haemophilus, Lautropia, Leptotrichia, Neisseria, Porphyromonas, Prevotella, Rothia* and *Veillonella* on genus level (Appendix A).

#### 2.1.1. Distribution of Microbial Genera in Different Oral Niches and Definition of Metaniches

Regarding the microbial composition of all seven niches for each healthy individual, no clustering was detectable for individual oral microbiotas. However, three distinct clusters were observed in a dendrogram based on generalized UniFrac distances, roughly correlating with groups of sampling sites (Figure 1). The first branch showed clustering of plaque (P) and gingival crevicular fluid (GCF), the second branch for saliva (S), tongue (T), and hard palate (HP), and the last branch showed clusters for cheek (C) and sublingual area (U) with some inter-individual variabilities. According to these branches, the frequencies of microbial genera showed specific patterns for P-GCF, S-T-HP, and C-U (Figure 1). Therefore, we could define three specific microbial “metaniches” according to the given data:(1)supragingival plaque and gingival crevicular fluid (P-GCF);(2)saliva-tongue-hard palate (S-T-HP);(3)cheek and sublingual area (C-U).

Most samples of metaniche P-GCF showed relatively higher proportions of the following genera compared to metaniches S-T-HP and C-U: *Actinomyes*, *Aggregatibacter*, *Fusobacterium, Leptotrichia, Capnocytophaga, Corynebacterium, Lautropia*, *Campylobacter* and *Tannerella*.

Compared to metaniches P-GCF and C-U, samples from branch S-T-HP showed higher proportions of the subsequent genera: *Prevotella, Neisseria*, *Veillonella, Porphyromonas*, *Granulicatella,* and *Alloprevotella*.

Regarding metaniche C-U compared to P-GCF and S-T-HP noticeably higher frequencies were seen for the following genera: *Haemophilus*, *Streptococcus*, *Gemella* and *Actinobacillus*.

Despite the recognizable differences between the three branches seen in the dendrogram (Figure 1), some genera showed similar amounts in all sites like *Rothia*.

According to the Shannon diversity index, samples of metaniche P-GCF showed a significantly higher alpha diversity compared to metaniche S-T-HP and C-U (3.64 ± 0.07 vs. 3.20 ± 0.05 vs. 2.58 ± 0.13, *p* < 0.0001), while metaniche C-U showed the lowest diversity (Figure 1, right side).

The grouping of niches into metaniches was confirmed by analysing beta diversity using multidimensional scaling (MDS) plots. We found the ellipse centres of plaque and gingival crevicular fluid, of cheek and sublingual area, and of saliva, tongue and hard palate, each in close proximity to each other (Figure 2a). While showing hardly any overlap to the other groups, the microbial profiles of P-GCF were in closest proximity to saliva (Figure 2a). In contrast, saliva, tongue, hard palate, cheek and sublingual area exhibited overlapping microbial profiles. This would make saliva to be the “connecting part” between all oral niches (Figure 2a). When grouped according to the identified clusters, the multidimensional scaling (MDS) plot underlined the presence of distinct metaniches in the oral cavity (Figure 2b).

To summarize, we could define three specific microbial metaniches according to the given data “P-GCF”, “S-T-HP” and “C-U”.

#### 2.1.2. Characteristic Species for Defined Metaniches

Characteristic microbial patterns for defined metaniches were not only seen on the genus level, but also on the OTU/species level. Analysis of statistically significant enrichment or depletion depending on the oral metaniche identified 24 OTUs (Figure 3).

Metaniche “P-GCF” demonstrated clearly higher values of the following OTUs compared to metaniche “C-U” and “S-T-HP”: **OTU_14** (*Rothia* sp.); **OTU_16** (*Campylobacter gracilis*); **OTU_20** (*Streptococcus sanguinis*); **OTU_24** (*Fusobacterium nucleatum*); **OTU_92** (*Capnocytophaga* sp.); **OTU_40028** (*Streptococcus* sp.)**; OTU_108** (*Prevotella oris*); **OTU_147** (*Leptotrichia buccalis*) and **OTU_231** (*Fusobacterium* sp.).

In **metaniche “S-T-HP”**, various OTUs were much more prevalent compared to metaniche “P-GCF” and “C-U”: **OTU_1** (*Streptococcus salivarius*); **OTU_2** (*Fusobacterium* (*pseudo*)*periodonticum*); **OTU_4** (*Prevotella melanogenica*); **OTU_6** (*Neisseria* sp.)**; OTU_12** (*Granulicatella adiacens*)**; OTU_15** (*Veillonella dispar*); **OTU_21** (*Actinomyces* sp.); **OTU_22** (*Streptococcus* sp.); **OTU_29** (*Porphyromonas* sp.); **OTU_39**
*(Saccharimonadaceae)*
**and OTU_45**
*(Campylobacter* sp.).

Concerning **metaniche “C-U”**, a smaller number of OTUs allowed discrimination from “S-T-HP” and “P-GCF”: **OTU_3** (*Haemophilus parainfluenzae*); **OTU_5** (*Streptococcus* sp.); **OTU_13**
*(Gemella haemolysans*) and **OTU_36** (*Haemophilus* sp.).

Taken together, species like *Rothia* sp., *Campylobacter gracilis*, *Streptococcus sanguinis*, *Fusobacterium nucleatum*, *Capnocytophaga* sp., *Streptococcus* sp. and *Prevotella oris* were characteristic for metaniche “**P-GCF**”, *Streptococcus salivarius*, *Fusobacterium* (*pseudo*)*periodonticum*, *Prevotella melanogenica*, *Granulicatella adiacens*, *Veillonella dispar*, *Actinomyces* sp., *Streptococcus* sp., *Porphyromonas* sp., *Saccharimonadaceae* and *Campylobacter* sp. were peculiar for metaniche “**S-T-HP**” and *Haemophilus parainfluenzae*, *Streptococcus* sp., *Gemella haemolysans* and *Haemophilus* sp. distinctive for metaniche “**C-U**”.

### 2.2. Cytokine Profile

In our first approach, we compared the overall concentration of cytokines in 140 samples of the 7 described oral niches. Subsequently, we analyzed whether each oral niche shows a characteristic pattern of cytokine expression and whether it is related to certain microbial metaniches.

#### 2.2.1. Cytokine Concentration in Different Oral Niches

The concentration (pg/mL) of granulocyte-macrophage colony-stimulating-factor (GM-CSF), IFN-γ, IL-2, IL-4, IL-6, IL-8 and IL-10 as well as TNF was measured in all oral niches (P, GCF, D, T, HP, C, U) and summed up (Appendix A). We observed significantly higher expression of cytokines in the gingival crevicular fluid with a mean value of 13,284.78 ± 2,869.64 pg/mL compared to the hard palate (292.40 ± 73.38 pg/mL, *p* < 0.0001), plaque (768.20 ± 182.90 pg/mL, *p* < 0.0001), saliva (241.70 ± 60.53 pg/mL, *p* < 0.0001), sublingual area (27.31 ± 7.05 pg/mL, *p* < 0.0001), cheek (61.48 ± 13.83 pg/mL, *p* < 0.0001) and tongue (201.50 ± 69.20 pg/mL, *p* < 0.0001), respectively. Plaque (768.20 ± 182.90 pg/mL) showed significantly higher values compared to hard palate (292.4 ± 73.38 pg/mL), tongue (201.50 ± 69.20 pg/mL) and saliva (292.40 ± 73.38), while the lowest concentrations were found in cheek (61.48 ± 13.83 pg/mL) and sublingual area (27.31 ± 7.05 pg/mL).

To conclude, the highest cytokine concentrations were found in the gingival crevicular fluid, while plaque also showed higher values. Saliva, tongue and hard palate showed similar median values, whereas cheek and sublingual area contained the lowest cytokine concentration. Notably, the cytokine pattern somewhat correlated with the microbial metaniches “P-GCF”, “S-T-HP” and “C-U”, while GCF had a uniquely high cytokine concentration. Therefore, GCF was excluded from the further comparison of the anti- and pro-inflammatory cytokines in the different oral sites.

Considering the anti-inflammatory cytokines IL-10 and IL-4 [19], highest concentrations were found in supragingival plaque (IL-10: 14.38 ± 2.53; IL-4: 0.47 ± 0.08; both in pg/mL), while all other niches present significantly lower values (saliva 0.72 ± 0.12 IL-10 and 0.19 ± 0.02 IL-4; tongue 0.53 ± 0.67 IL-10 and 0.10 ± 0.04 IL-4; hard palate 0.28 ± 0.15 IL-10 and 0.14 ± 0.07 IL-4; cheek 0.30 ± 0.05 IL-10 and 0.07 ± 0.02 IL-4; sublingual area 0.39 ± 0.13 IL-10 and 0.22 ± 0.09 IL-4; all in pg/mL).

Regarding the measured pro-inflammatory cytokines [19], we found the highest concentrations of GM-CSF, IFN-γ, IL-6/-8 and TNF in supragingival plaque (3.62 ± 0.84 GM-CSF; 38.08 ± 8.14 IFN-**γ**; 17.87 ± 4.31 IL-6; 5,979 ± 1,072 IL-8; 11.03 ± 1.63 TNF; all in pg/mL). Notably, we detected the highest expression of IL-1β in the hard palate (1,533 ± 456.70 pg/mL) and the highest values of IL-2 in the sublingual area (2.28 ± 0.98 pg/mL).

In accordance with the microbial metaniche “S-T-HP” similar cytokine concentrations were found in saliva, tongue and hard palate for the following cytokines:

GM-CSF: saliva 1.27 ± 0.59, tongue 0.99 ± 0.29, hard palate 0.97 ± 0.46;

IFN-γ: saliva 6.45 ± 0.55, tongue 5.27 ± 2.04, hard palate 9.39 ± 4.05;

IL-6: saliva 5.54 ± 1.42, tongue 10.44 ± 6.05, hard palate 8.59 ± 2.50 (all in pg/mL).

Fitting with the microbial metaniche “C-U” similar and overall lowest cytokine concentrations were seen in cheek and sublingual area for the following cytokines:

GM-CSF: cheek 0.49 ± 0.12, sublingual area 0.71 ± 0.27;

IL-1β: cheek 38.49 ± 8.24, sublingual area 10.28 ± 2.27;

IL-6: cheek 4.17 ± 0.93, sublingual area 2.46 ± 0.69;

IL-8: cheek 501 ± 68.16, sublingual area 216.90 ± 45.16;

TNF: cheek 1.72 ± 0.26, sublingual area 2.28 ± 0.53 (all in pg/mL).

In summary, gingival crevicular fluid presented the highest values of all measured cytokines compared to other niches, while higher values were also found in plaque for most measured anti- and proinflammatory cytokines (GM-CSF, IFN-γ, IL-4/-6/-8/-10). Conforming to the microbial metaniche “S-T-HP” saliva, tongue and hard palate showed similar amounts of GM-CSF, IFN-γ and IL-6. In accordance with the microbial metaniche “C-U” cheek and sublingual area presented similar and the overall lowest cytokine concentrations of GM-CSF, IL-1β/-6/-8 and TNF.

#### 2.2.2. Characteristic Cytokine Expression Pattern

In a second approach, we analyzed whether metaniches as seen in the microbial patterns also exist with respect to the concentrations and relative quantities of the measured cytokines at the different sites of sampling (Figure 4 and Figure 5).

Across all oral niches, the most dominant cytokine was IL-8, while IL-1β was also highly prevalent in most sites (Figure 4). Regarding IL-8 and IL-1β, similarities between hard palate and tongue were found with comparable concentrations of IL-1β and IL-8 showing similarities to the microbial metaniche “saliva-tongue-hard palate” (S-T-HP). Cheek and sublingual area also presented the lowest overall concentrations of IL-1β and IL-8 with similarities to the microbial metaniche “cheek-sublingual area” (C-U).

Regarding the other measured cytokines, similarities were also found for the immunological metaniche supragingival plaque and gingival crevicular fluid (P-GCF) with high concentrations of IL-10, IFN-γ, IL-6 and TNF and low concentrations of GM-CSF, IL-2 and IL-4 (Figure 4a,b). An “immunological metaniche” was also formed by tongue and hard palate (T-HP without S), which showed similar cytokine patterns with medium concentrations of IFN-γ, IL-6 and TNF and very low values of GM-CSF, IL-10, IL-2 and IL-4 (Figure 4d,e). Saliva took an intermediate position between the immunological metaniches “P-GCF”, “T-HP” and “C-U”, with high concentrations of IFN-γ, IL-6 and TNF similar to “P-GCF”, but barely detectable IL-10 as observed in “T-HP” and “C-U”, which are also characterized by relatively low IL-10 values (Figure 4a–f). Moreover, some samples of saliva showed high concentrations of GM-CSF and IL-2, which was not seen in other niches (Figure 4c). Cheek and sublingual area presented varying concentrations of IFN-γ, IL-6 and TNF with very low values of GM-CSF, IL-10, IL-2 and IL-4. However, a definitive correlation was not seen probably due to the low absolute cytokine concentrations in the sublingual area (Figure 4f,g).

In summary, by analyzing the relative quantities of cytokines measured in the defined microbial metaniches, we discovered characteristic cytokine profiles (Figure 5): Metaniche P-GCF is dominated by IL-8 with a low relative distribution of IL-1β, while all other measured cytokines ranged below 0.04%. Compared to P-GCF and C-U, metaniche S-T-HP showed a high proportion of IL-1β, being the second most abundant cytokine after IL-8. All other measured cytokines were also below 0.04%. While metaniche C-U was also dominated by IL-8, it was interesting to see that not only IL-1β but also IFN-ɣ, IL-6 and TNF showed relatively high values, whereas the values for IL-2, GM-CSF, IL-10 and IL-4 were again below 0.04%.

### 2.3. Correlation Between the Microbial Composition and Immunological Profile

In a final analysis, we investigated a possible association between the microbial composition and the immunological profiles of the defined metaniches. For that purpose, relative abundances of genera were correlated with cytokine levels (Figure 6). 

The following correlations between the cytokine profiles and the defined microbial metaniches were seen:

Regarding metaniche “P-GCF”, positive correlations with high to highest support were found between cytokine levels and relative amounts of the genera *Aggregatibacter*, *Fusobacterium*, *Gemella* and *Streptococcus*, while negative correlations were seen with *Tannerella*, *Leptotrichia*, *Corynebacterium*, *Capnocytophaga*, *Saccharimononadae* and *Neisseriacae* (Figure 6a). In the metaniche “S-T-HP”, positive correlations with the highest support were found for *Gemella* spp., while variable positive connections were identified for the genera *Capnocytophaga*, *Streptococcus*, *Porphyromonas* and *Campylobacter* (Figure 6b). Cytokine levels negatively correlated with relative amounts of *Prevotella*, *Veillonella* and *Actinomyces* species (Figure 6b). In the metaniche “C-U”, we identified positive correlations with high to highest support for the genera *Campylobacter*, *Granulicatella* and *Prevotella*, while *Capnocytophaga*, *Actinobacillus* and *Leptotrichia* sp. negatively correlated with cytokine levels (Figure 6c).

## 3. Discussion

Regarding the pathogenesis of periodontal disease, previous theories relied on culture-based methods: Socransky et al. [29] divided bacterial species according to their pathogenicity into five microbial complexes and regarded the presence of the “red complex” consisting of *Porphyromonas gingivalis*, *Tannerella forsythia* and *Treponema denticola* as highly associated with periodontal inflammation. However, as mentioned above a new explanatory model defined periodontal health as a state of “symbiosis” and periodontal disease as a “dysbiotic inflammatory disease” [2]. To characterize the oral microbiota to its full complexity, the sequencing of the 16S rRNA genes has become an acknowledged, culture-independent method [1,3,4]. As the oral cavity contains several anatomically diverse oral niches, characterisation of the oral microbiota should not only focus on its whole complexity but also on their distinct niches.

### 3.1. Microbial Composition in Different Oral Niches

The first aim of this study was to investigate the microbial composition in different oral niches of young adults with excellent periodontal health and oral hygiene. To this aim, we analyzed the microbial composition in samples gained from 7 distinct oral niches (plaque (P), gingival crevicular fluid (GCF), saliva (S), tongue (T), hard palate (HP), cheek (C) and sublingual area (U)) using next generating sequencing.

We detected differences in the relative abundances of microbial genera between different oral niches. Moreover, with cluster analyses of the microbial composition of individual samples, we could define three specific microbial “metaniches”: “P-GCF”, “S-T-HP” and “C-U”. A previous report also observed the formation of groups between seven diverse habitats of the digestive tract, which partly matched our results [30]. Here, four groups were defined: (1) buccal mucosa, keratinized gingiva and hard palate; (2) saliva, tongue, tonsils and throat; (3) sub- and supra-gingival plaques; (4) stool. In analogy to our results, clustering between saliva and tongue and differences in the microbial compositions between buccal mucosa, saliva/tongue and plaque were observed [30]. However, in contrast to our results, buccal mucosa and hard palate showed distinct clustering [30].

Regarding the microbial composition in different oral niches, another previous study analyzed the tongue dorsum, hard palate, buccal mucosa, keratinized gingiva, supragingival and subgingival plaque, and saliva with or without rinsing in 20 healthy subjects and demonstrated that no significant differences were seen between the study participants [4]. Consistent with this publication, no subject-related clustering was observed in our study. However, other previous studies detected not only site-specific but also subject-specific bacterial species in a study population of 5 subjects with a broad range of age (23 to 55 years) [17]. This dissimilarity might be explained by age-related alterations of the oral microbiota leading to “subject”-specific bacterial compositions. Also in line with our results, another investigation presented a “core microbiota” in health with strong similarities between unrelated individuals and pointed out that niche-specific microbiotas might exist [16]. However, only five niches (dental surfaces, cheek, hard palate, tongue and saliva) from three subjects were analyzed.

The similarities between plaque and gingival crevicular fluid and the clustering into metaniche “P-GCF” might be explained by their anatomical proximity to dental structures and direct contact to the gingival margin. Plaque can be found on the supra- or subgingival surface of teeth as a structurally and functionally organized complex microbial community with specific formation stages and has been referred to as “dental biofilm” [31]. This biofilm is considered the primary etiologic factor for polymicrobial inflammatory disease [32]. It was shown that the presence of a plaque/dental biofilm induces the production of the gingival crevicular fluid [33], which is a physiological fluid originating from blood vessels in the gingiva underlying the epithelium lining of the gingival sulcus [20,21]. Moreover, GCF serves as nutrition for biofilms in gingivitis and periodontitis and sulcus fluid flow rate (SFFR) increases dependent on the quantity and diversity of dental biofilm [34].

The origin and distribution of the saliva and the “salivary pellicles” found on tongue, hard palate, cheek and sublingual area are likely to account for the microbial similarities seen between “S-T-HP” and “C-U”. Salivary glands can be differentiated into “major” and “minor” glands with varying quality and quantity of saliva [35]. Major glands are located in the buccal cheek (region of the first upper molar) and the sublingual area [35], which might explain the correlation seen between “C” and “U”. Minor glands are found in the tongue, palate, lower lip [35], which might explain the microbial relationship between “T” and “HP”. Considering anatomical variations, salivary flow is not constant, with the highest flow rate in the sublingual area and lowest in the anterior region of the maxillary front teeth [35]. The high clearance rate in the sublingual area and cheek with a constant salivary flow from major glands offers a plausible explanation for the least microbial diversity found in the metaniche “C-U”.

Moreover, the close microbial relationship between saliva, tongue and hard palate and their clustering into metaniche “S-T-HP” might result from the functional tasks of hard palate and tongue, which are in close contact almost constantly.

In our study, most samples of the metaniche P-GCF comprised higher fractions of *Actinomyces, Aggregatibacter, Fusobacterium, Leptotrichia, Capnocytophaga, Corynebacterium, Lautropia, Campylobacter* and *Tannerella* and exhibited a high alpha diversity. Consistent with our results, a previous study demonstrated that plaque showed a high diversity including *Actinomyces, Fusobacterium*, *Corynebacterium*, *Capnocytophaga* and that saliva as well as tongue showed a high prevalence of Prevotella and *Neisseria* [30]. Partly similar results for plaque were found in a later consensus report, which indicated that the dental biofilm in periodontal health is dominated by the genera *Neisseria, Streptococcus, Actinomyces, Veillonella* and *Granulicatella*, but shifts to a more diverse and higher specialized form in gingivitis and periodontitis [34]. Samples from the metaniche S-T-HP showed a relatively high abundance of *Prevotella, Neisseria, Veillonella, Porphyromonas, Granulicatella* and *Alloprevotella*. Findings were partly similar to a previous report demonstrating the presence of *Veillonella, Prevotella, Neisseria* and *Granulicatella* in the hard palate and tongue dorsum. However, the highest values were found for *Streptococcus* [17]. The metaniche C-U presented a high relative abundance of *Streptococcus, Haemophilus, Gemella* and *Actinobacillus* with a very high abundance of *Streptococcus*, which was consistent with a previous study indicating that *Streptococcus* was dominant in the cheek [30].

On the species level, Campylobacter gracilis, Streptococcus sanguinis, Fusobacterium nucleatum, Capnocytophaga and Prevotella oris were characteristic for the metaniche “P-GCF”, Streptococcus salivarius, Fusobacterium (pseudo)periodonticum, Prevotella melanogenica, Granulicatella adiacens, Veillonella dispar and Saccharimonadaceae were typical for “S-T-HP”, and Haemophilus parainfluenzae and Gemella haemolysans signified “C-U”. Similar findings were reported in a previous study [17], e.g., for (1) tooth surface/plaque (Streptococcus sanguinis); (2) tongue dorsum (Streptococcus salivarius); and (3) cheek (Gemella haemolysans). Our results are also in accordance with another recent publication demonstrating that the microbial diversity was significantly different between oral niches [4]; however, in this study, characteristic OTUs were only analyzed in each oral niche without grouping into metaniches.

### 3.2. Cytokine Expression in Different Oral Niches

Pathogenesis of periodontitis includes a complex interaction between microbial communities and the host immune response, resulting in the production of inflammatory mediators that alter connective tissue and bone metabolism [2,19,36]. Cytokines are soluble proteins that bind to specific receptors on target cells, activate intracellular signalling cascades and play important roles in acute-phase response, a primary defence reaction that protects the body against bacterial products in periodontal tissue breakdown [22,28,37]. In the diagnosis of periodontal disease, an increase in the amount of GCF and the presence of certain pro-inflammatory cytokines such as IL-1β, IL-6, IL-8, IFN-ɣ and TNF has been correlated with the progression of periodontitis, while pro-inflammatory cytokine levels decreased after periodontal therapy [19,23,24,38,39,40]. While cytokine expression in GCF has been investigated in periodontal disease [19], it has rarely been analyzed in other oral niches like the salivary pellicle on different soft tissue surfaces and has not yet been related to the oral microbiota.

In our study, we focused on the expression patterns of nine different cytokines, including the pro-inflammatory cytokines IL-1β, IL-2, IL-6, TNF and IFN-γ, the chemokine IL-8 and the anti-inflammatory mediators IL-4 and IL-10 [19]. Therefore, we performed a broad analysis in 20 age-matched young adults with excellent oral health (10 males and 10 females) providing characterisation of cytokine profiles of the aforementioned oral niches.

Regarding the microbial metaniche “P-GCF”, similarities were also found for the “immunological metaniche” supragingival plaque and gingival crevicular fluid: First of all, cytokine concentrations were significantly higher in the gingival crevicular fluid compared to all other niches followed by supragingival plaque which was found to be significantly higher compared to saliva, tongue, hard palate, cheek and sublingual area. Very high concentrations of IL-8, high concentrations of IL-10, IFN-γ, IL-6, TNF and low concentrations of GM-CSF, IL-2 and IL-4 were detected in “P-GCF”. The high cytokine concentration in GCF might originate from the enclosed area in the gingival sulcus, which is protected from salivary clearance and dilution effects. GCF is synthesized in the connective tissue, which is part of the dental attachment and secretes directly into the gingival sulcus where it can be collected [19]. In periodontal health, the sulcus fluid flow rate is low and GCF is described as a transudate following an osmotic gradient, while in periodontitis, GCF is comparable to an inflammatory exudate showing high sulcus fluid flow rates [20]. Moreover, GCF serves as a transport medium carrying nutrients for subgingival bacteria and several immune cells like neutrophils, which are the most common leukocytes recruited in periodontal disease [41]. Anatomically, plaque accumulation is often located on the gingival margin inducing the production of the gingival crevicular fluid [33], which might be the reason for the second-highest cytokine concentrations as part of the immune response. Bacterial components found in supragingival plaque, e.g., lipopolysaccharides, stimulate various inflammatory mediators and immune cells found in the connective tissue [19]. Initially, mast cells produce TNF triggering vasodilatation and the recruitment of polymorphonuclear leukocytes (PMNs), which can induce tissue destruction in periodontitis by the release of lysosomal enzymes [19]. Simultaneously, resident cells in the connective tissue, which are mainly gingival fibroblasts, release pro-inflammatory cytokines like IFN-γ, IL-1β, IL-6, IL-8 and PGE_2_ activating a cascade of events in the host response [19]. Moreover, antigen-presenting cells stimulate differentiation of naïve T cells into various cell types, which release different cytokines like type 1 T helper (Th1) cells (production of IFN-γ, IL-2 and TNF), Th2 cells (e.g., IL-4, IL-10, IL-13), regulatory T cells (e.g., IL-10) and Th17 cells (e.g., IL-6, TNF- and IL-17) [19].

Considering the percentage of cytokine expression in all niches, IL-8 was found to be the most dominant cytokine. Similar to our results, high IL-8 levels were previously described in GCF of healthy individuals [42]. We observed similarities in the cytokine expression pattern between hard palate and tongue, with a relatively low percentage of IL-8 (<75%) and a high percentage of IL-1β (>20%), which might result from similar fluids produced by minor glands found in hard palate and tongue [35]. Cheek and sublingual area presented lowest cytokine concentrations, which matches the lowest microbial diversity found in metaniche “C-U” and can also be explained by the highest salivary flow due to the presence of salivary glands in those niches [35]. The highest relative quantities of IFN-γ, IL-6, GM-CSF, and IL-2 were found in the sublingual area and the cheek, indicating analogies between the cheek and sublingual area and the presence of an immunological metaniche “C-U”.

Cytokine concentrations in saliva suggested similarities to “P-GCF” but also to “T-HP” and “C-U”. On the one hand, similarities of tongue/hard palate and cheek/sublingual area with the salivary compartment probably originate from the formation of the “salivary pellicle”, which builds a thin conditioning layer on all soft tissues in the oral cavity [43,44,45]. As mentioned before, saliva is mainly produced by salivary glands found not only in the cheek and in the sublingual area but also in the hard palate and tongue [35]. Saliva also contains gingival crevicular fluid [35], which is crucial in antimicrobial defence and maintenance of periodontal health by flushing microbial components and bacteria out of the gingival sulcus [46]. Taken together, it is plausible that cytokine concentrations in saliva show properties of all oral niches.

### 3.3. Correlation between Microbiota Composition and Cytokine Expression 

Higher pro-inflammatory cytokine levels and a change in the microbial composition have not only been detected in periodontal disease, but have also been connected to other oral conditions. Regular smoking, which is a predisposing factor for oral and lung cancer, correlated with higher levels of the proinflammatory cytokine IL-2 and with a higher microbial prevalence of *Proteobacteria*, *Firmicutes, Bacteroidetes*, *Fusobacteria* and *Actinobacteria* [26]. Moreover, there is evidence that *Porphyromonas gingivalis* might play an important role in the development of oral squamous cell carcinoma [47]. Therefore, salivary biomarkers like the pro-inflammatory cytokines IL-1β, IL-2 and TNF might be used as a future diagnostic tool for oral squamous cell carcinoma [27].

Notably, the present investigation in healthy volunteers revealed positive correlations between increased relative quantities of cytokines (GM-CSF, IFN-γ, IL-10, IL-1β, IL-2, IL-4, IL-6, IL-8 and TNF) and specific microbial genera within the defined metaniches: Positive correlations with *Aggregatibacter* and *Fusobacterium* were characteristic for the metaniche “P-GCF”, which are typical commensals of the dental biofilm. These associations might result from the properties of *Fusobacterium nucleatum*, which can stimulate the production of IL-6 and IL-8 in the gingiva [48], and the immunoregulatory potential of *Aggregatibacter actinomycetemcomitans* inducing expression of IFN-γ [49,50]. A moderate up-regulation of these cytokines might help in the defence of pathogens and the maintenance of “symbiosis” between microbial players and the immunity of the host, while high and uncontrolled up-regulation correlated with periodontitis [19]. In contrast, dysbiosis and inflammation mutually amplify each other in periodontitis: the innate immune system responds to colonization by pivotal pathogens (e.g., *P. gingivalis*) with dysregulated and increased production of GCF, which contains specific molecules from periodontal tissue destruction, and finally inhibits the growth of eubiotic species and fosters the growth of periodontitis associated taxa [41,51].

Positive correlations were also seen for *Gemella* and *Capnocytophaga* in the metaniche “S-T-HP” and for *Campylobacter* and *Granulicatella* in the metaniche “C-U”. The genera *Campylobacter*, *Capnocytophaga* and *Gemella* were previously associated with periodontal health [52]. The genus *Granulicatella* is also a common and frequent component of healthy oral microbiotas [53]. However, little is known to date about the aforementioned microbes in the oral context and possible immune-stimulating properties.

Regarding associations between microbial composition and immunological features in different oral niches, only a small amount of data are available for periodontal health except comparisons with oral inflammatory disease: Higher levels of pro-inflammatory cytokines like IL-6 and TNF in GCF have been linked to the presence of “red complex” bacteria in supragingival plaque [54] or other periodontal pathogens in deep periodontal pockets [55]. However, to our best knowledge, no other studies are available that have investigated more than two oral niches with regard to their microbial composition and cytokine profile and/or correlations between immune response and bacterial composition. It was shown that tongue cleaning, which might lead to a reduction of microbial colonization, correlated with decreased levels of IL-1β and IL-8 in GCF in patients with gingivitis [56], suggesting that “health” of one oral niche can even lead to less inflammation in other oral niches. In line with these findings, we detected similarities in the oral microbiota and cytokine profile of anatomical adjunct niches or niches in direct contact facilitating the interaction between them.

## 4. Materials and Methods

The study was designed as a prospective, exploratory observational clinical trial. The study was performed in accordance with the Helsinki protocol and the local ethics committee of the Friedrich-Alexander-University Erlangen-Nürnberg (Krankenhausstraße 12, 91054 Erlangen, Vote number: 111_20 B). After written informed consent and fulfilment of the inclusion criteria, study participants were enrolled in the study. The inclusion and exclusion criteria are given in Table 1.

### 4.1. Study Population

Twenty healthy study participants (10 males, 10 females) with a mean age of 27.20 years (range 23 to 32) fulfilling the inclusion and not meeting the exclusion criteria given in Table 1 were enrolled in the study. To ensure excellent oral hygiene, all study participants were either dentistry students, dentists, orthodontists or scientific assistants at the university hospital of Erlangen-Nürnberg. The mean number of teeth was 29 (range 26–32) without apparent mobility (Grade 0) (Appendix A). The periodontal probing depth (PPD) was measured with a periodontal probe (WHO probe) on six sites per tooth and ranged from 1.4 mm to 2.3 mm with a mean value of 1.8 mm without signs of pathological pockets. Periodontal and gingival health were evaluated with the bleeding on probing index (BOP), which was measured at six sites per tooth. The average BOP was 0.7%. In accordance with the definition of periodontal health [57], all subjects presented excellent periodontal health. For determination of the approximal plaque index (API) the Mira-2-Ton® disclosing agent was used, which is a two-tone erythrosine-free disclosing dye solution Mira-2-Ton® (Hager & Werken, Duisburg, Germany) that can help to distinguish blue-dyed older plaque from pink-dyed newer plaque. The mean API was 13% (range 7–18%) representing in only pink-dyed newer plaque as the old blue-dyed plaque was not detectable. In accordance with the definition of good oral hygiene with API < 25% [58,59], all subjects presented excellent oral hygiene. All subjects used a powered toothbrush and additionally interproximal hygiene devices. Eleven subjects used dental floss regularly, whereas nine subjects used dental floss only on an occasional basis. All participants are part of a dental recall system, which includes regular professional prophylaxis at a dental practice or dental clinic.

### 4.2. Sample Collection

After obtaining informed consent, 280 samples were collected from 7 different oral niches (2 samples per niche) of each study participant to analyse the composition of the oral microbiota and the cytokine expression in each oral niche. Individuals were instructed not to eat or drink anything except water up to three hours before saliva collection. Unstimulated saliva samples were gained by the spitting method and collected in microtubes. After collection of saliva, soft tissue samples were collected in sterile tubes from the hard palate (middle part), the tongue (middle and anterior part), the sublingual area (middle part) and the cheek (right side, molar region), using sterile swabs wiping over the area several times for about 30 s. Moreover, hard tissue samples of supragingival plaque were taken by sterile curettes and wiped off using sterile swabs. Samples of gingival sulcus fluid were collected from 6 periodontal pockets of the mesiobuccal side of defined teeth (Ramfjord teeth: 16, 21, 24, 36, 41, 44) using sterile paper strips. The samples were collected using a standardized protocol, which was modified to prior described protocols [60,61]: draining with cotton rolls, careful air-drying of the area, mild insertion of paper strip (ProFlow, Amityville, NY, USA) in the gingival sulcus for 30 s in the premolar and molar region and for 60 s in the incisor region (as study population consisted of healthy individuals, insertion for 30 s in the incisor region was not suitable for the collection of adequate amounts of sulcus fluid). After calibration of the Periotron 8000 before each usage, measurement of the amount of sulcus fluid flow rate (SFFR) were conducted subsequently for one sample each niche using Periotron 8000 according to the manufacturer’s instructions to determine the cytokine concentration. To ensure that paper strips and oral sterile culture swabs were free from bacterial DNA, all materials were tested in advance.

After collection of samples and immediate storage on dry ice, clinical measurements were performed on all study participants using a standardized working process and diagnosis sheet: dental notation including the number of teeth (missing teeth due to extraction or agnesia), full mouth probing of periodontal pockets (periodontal probing depth, PPD), bleeding-on-probing (BOP)-positive sites and approximal plaque index (API) were measured and calculated.

Patient recruitment and sample collection were performed at the department of orthodontics and orofacial orthopedics of the university hospital of Erlangen. Storage of samples as well as isolation of bacterial DNA was performed in the research laboratory of the department of orthodontics and orofacial orthopedics.

### 4.3. Determination of Cytokine Concentrations

For determination of cytokine concentrations, collected samples were isolated from swabs by centrifugation (1 min at 23,000× *g* at 4 °C), volume was measured and samples were diluted with diluent 43 (Mesoscale Discovery, R50AG-2) to a volume of >100 µL. Saliva samples were directly diluted 1:3 with diluent 43. For recovery of cytokines from crevicular samples, sterile paper stripes of each test person were sequentially shaken in 150 µL diluent 43 at 4 °C for 5 min each. Residual volume was measured. The dilution factor was calculated by dividing the residual volume by the total volume of crevicular fluid as determined by Periotron measurements. Plaque samples were suspended in 125 µL diluent 43. As weighing of plaque samples was not feasible due to variation of test tube weight, we assigned these samples into three groups according to their pellet size after centrifugation of diluted suspensions: large, medium size and small plaque samples, corresponding 8 µL, 4 µL or 2 µL, respectively. Dilution factors for plaque samples were calculated from these assignments.

Cytokine concentrations were determined by multiplex immunoassay in 96 well plates with a U-PLEX Biomarker Group 1 (hu) assay (Mesoscale discovery; K15067L-2) on a MESO QuickPlex SQ 120 instrument (Mesoscale discovery). The assay was performed according to the manufacturer’s instructions, standard dilution series (one per each plate) and samples were measured in duplicates.

### 4.4. Microbiota Analysis

DNA of samples was extracted with a ZymoBIOMICS DNA Microprep Kit (Zymo Research, Freiburg, Germany) in combination with a BeadRupter 24 (Omni International, Kennesaw, GA, USA). The bacterial 16S rRNA gene was amplified with primers 27Fmod 5′-AGRGTTTGATCMTGGCTCAG-3′ and 519R 5′-GWATTACCGCGGCKGCTG-3′ [62]. A total of 12 primer pairs, each modified with different 0-11 bp heterogeneity spacers and adapters for the Illumina Nextera XT system were synthesized (IDT, Munich, Germany) and used to amplify the 16S V1-V3 region. For PCR the KAPA HiFi HotStart ReadyMix (Roche, Mannheim, Germany) was used in a Mastercycler nexus Gradient (Eppendorf, Hamburg, Germany) with the following program: 3′ 95 °C, 25 cycles of 30″ 95 °C, 30″ 55 °C, 35″ 72 °C and finally 5′ 95 °C. Fragments were purified with AMPure XP beads (Beckman-Coulter, Krefeld, Germany). A second indexing PCR was performed (8 cycles) using the Nextera Index Kit v2 Sets A and B (Illumina, Munich, Germany). After final purification (AMPure XP), equal molarities of the samples were pooled and subjected to 2 × 300 bp paired-end sequencing on a MiSeq system (Illumina).

Sequence data were demultiplexed according to the index sequences with the “Generate Fastq” workflow in MiSeq Reporter (Illumina). Sequencing files in the fastq format are available through the National Center for Biotechnology Information (NCBI) Sequence Read Archive (SRA) under BioProject PRJNA668021. Sequences were filtered based on quality using Trimmomatic v0.39 [63] and primer sequences were removed with Cutadapt v2.9 [64]. For OTU-based classification with the USEARCH v8 algorithm [65] sequences were re-multiplexed using a Perl script (I. Lagkouvardos, available from www.imngs.org) and submitted to the IMNGS server [66]. We obtained in median 45,338 OTU counts per sample (range 2334–112,749). Taxonomic classification of the identified OTUs was done using the SILVA alignment, classification and tree service [67] with SILVA database release 138.1 and NCBI nucleotide BLAST [68]. All further analyses were performed using the R-based pipeline “Rhea” [69] and ggplot2 [70] (for visualization) within R v4.0.2 [71].

### 4.5. Statistics

The statistical analysis of cytokine concentrations was done with Microsoft Excel 365 (Microsoft, Redmond, WA, USA), R v4.0.2 [71] and GraphPad Prism 8 statistical software (GraphPad Software, San Diego, CA, USA). If not stated otherwise, data are presented as mean ± Standard Error of Mean (SEM) and fraction of total. The data sets were analyzed by using the Kruskal–Wallis test and Dunn’s multiple comparisons test. Differences were considered significant with *p*-values < 0.05.

## 5. Conclusions

The presented investigations allowed the definition of three distinct metaniches: (1) plaque and gingival crevicular fluid; (2) saliva, tongue and hard palate; and (3) cheek and sublingual area according to their microbial composition. These metaniches were associated with defined cytokine profiles, except saliva, which showed relationships to the cytokine profile of all oral niches. These defined metaniches provide new insights into microenvironmental similarities between anatomically diverse oral niches and indicate possible interactions between niches and their defined metaniches. The metaniches in the oral cavity are formed and influenced by (1) the anatomical proximity and interactions in host defence (dental biofilm can be found on the gingival margin and stimulates the production of gingival crevicular fluid leading to flushing out of bacterial products); (2) the function (permanent contact between tongue and saliva—also during swallowing and speech); (3) the salivary glands and flow rate of saliva (major glands found in the cheek and sublingual area with highest salivary flow, while minor glands are present in hard palate and tongue); and (4) the mixed composition of saliva (fluids from salivary glands found in diverse oral niches and GCF), building a “salivary pellicle” on hard and soft tissues. The understanding of these metaniches is crucial when analysing changes in the composition of the oral microbiota or immune responses associated with treatment or oral diseases. Hence, future studies are necessary to improve our knowledge of site-specific alterations, e.g., in periodontitis. This knowledge might help to define the state of “dysbiosis” more precisely and detect future biomarkers in periodontal disease. It might also help to improve potential site-specific preventive or therapeutic strategies, which not only have an impact on the directly treated oral niche but also associated or distant niche(s).

## Figures and Tables

**Figure 1 ijms-21-08218-f001:**
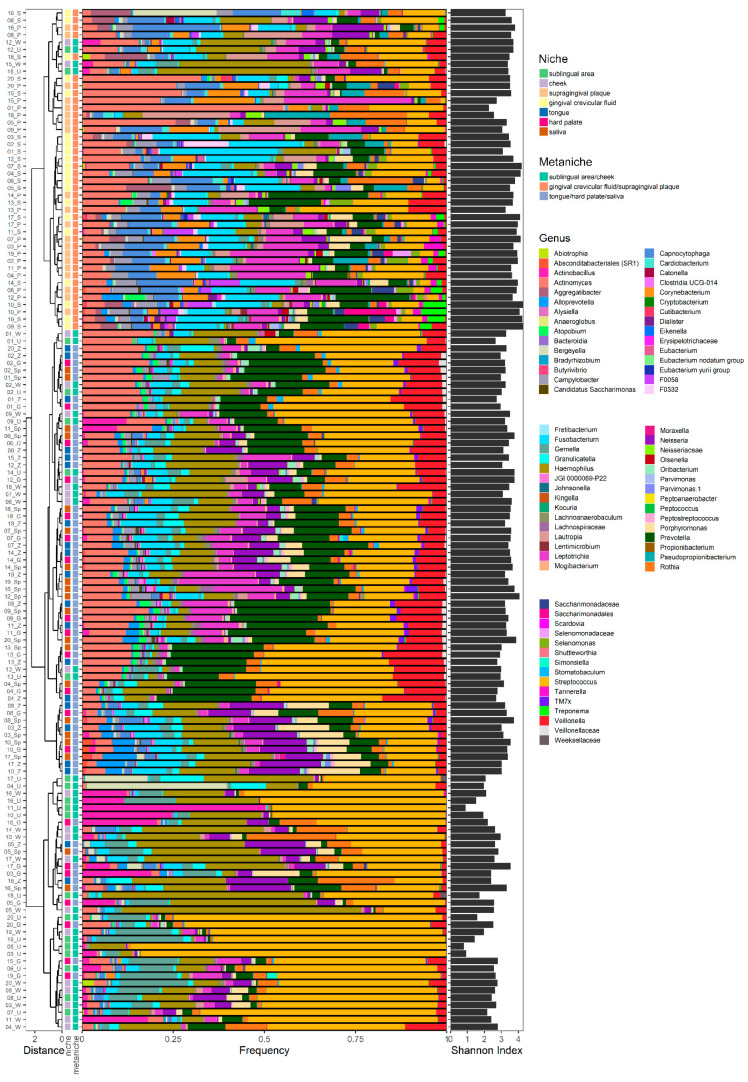
Dendrogram, genus frequencies plot, and Shannon index representing the genera detected in different oral niches and metaniches. A dendrogram based on generalized UniFrac distances is given for all oral niches (sublingual area, cheek, supragingival plaque, gingival crevicular fluid, tongue, hard palate, saliva) and metaniches (sublingual area/cheek; gingival crevicular fluid/supragingival plaque; tongue/hard palate/saliva) for each individual subject and sample (colour key is given on the right side). Genus frequencies found in the individual microbiotas are given with a colour key for the different genera, which are sorted alphabetically and presented horizontally from the left to the right side. The Shannon diversity index is given on the right side.

**Figure 2 ijms-21-08218-f002:**
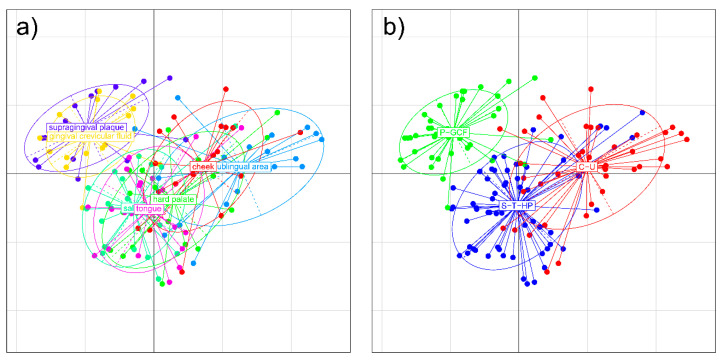
MDS plots of microbial profiles are given. (**a**) MDS plot for microbial profiles for all oral niches (plaque, gingival crevicular fluid, saliva, tongue, palate, cheek, sublingual) are displayed; (**b**) MDS plot of microbial profiles for defined metaniches (P-GCF = plaque/gingival crevicular fluid, S-T-HP = saliva/tongue/palate, C-U = cheek/sublingual) are displayed.

**Figure 3 ijms-21-08218-f003:**
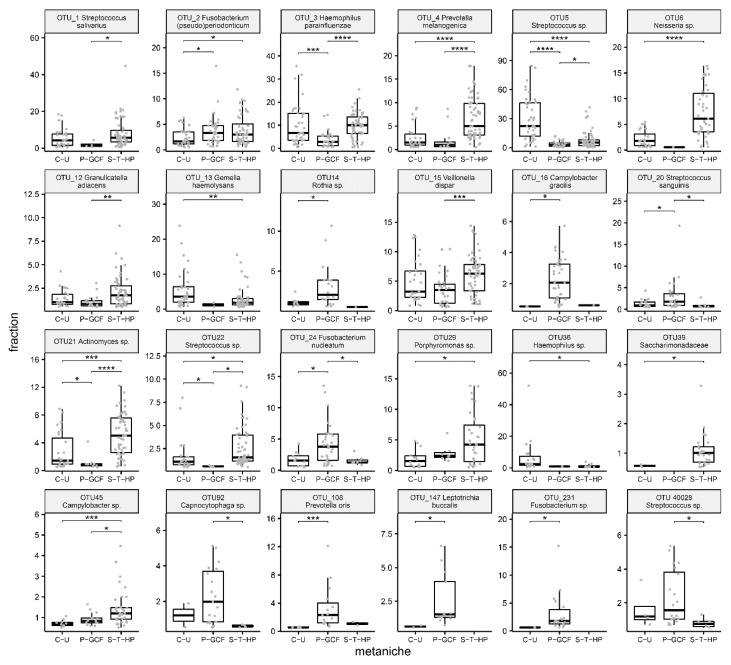
Characteristic operational taxonomic units (OTUs) for three different defined metaniches “plaque and gingival crevicular fluid” (P-GCF), “saliva, tongue and hard palate” (S-T-HP) and “cheek-sublingual” (C-U) are shown. The data is presented as median, 25 and 75 percentiles (box) and 1.5 × interquartile range of the 25 and 75 percentiles (whiskers). Raw data (*n* = 0-59) is displayed as overlaying grey points. Statistical analysis was performed using Mann-Whitney U-tests and was adjusted for sample size using the Benjamin–Hochberg algorithm. * 0.01 ≤ *p* < 0.05, ** 0.001 ≤ *p* < 0.01, *** 0.0001 ≤ *p* < 0.001, **** *p* < 0.0001.

**Figure 4 ijms-21-08218-f004:**
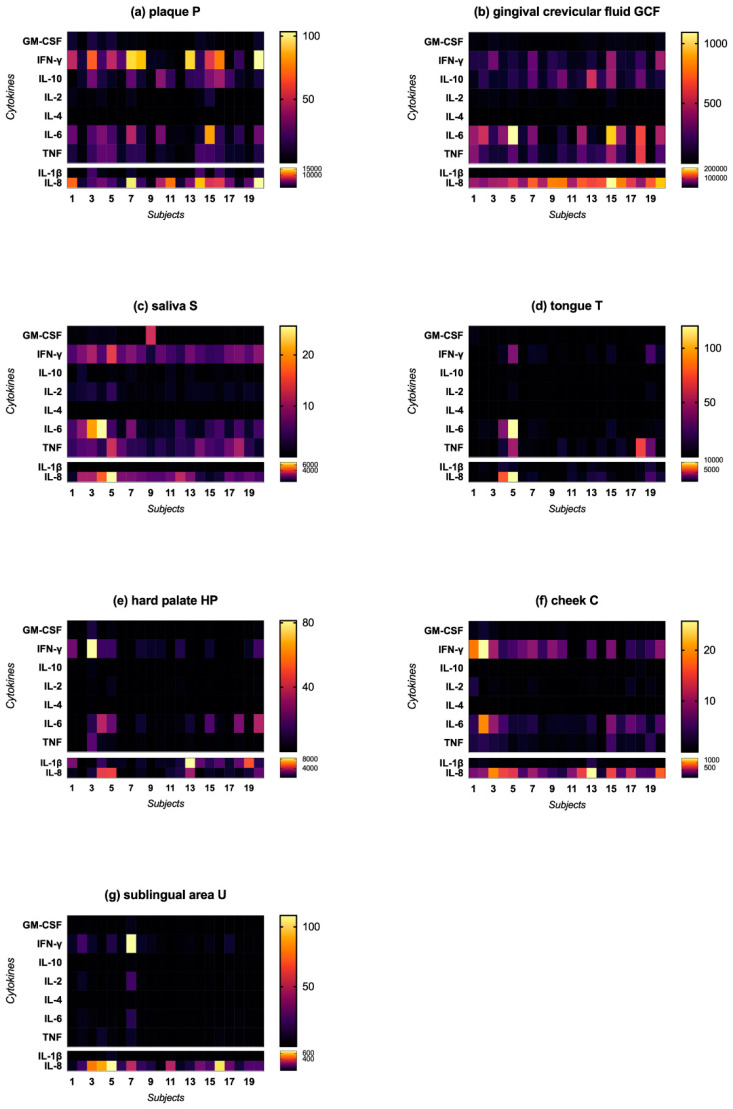
Heat map visualization of the cytokine concentration in different analyzed oral cavity sites including (**a**) supragingival plaque P, (**b**) gingival crevicular fluid GCF, (**c**) saliva S, (**d**) tongue T, (**e**) hard palate HP, (**f**) cheek C and (**g**) sublingual area U with the expression pattern of the pro- and anti-inflammatory cytokines. The cytokine concentration of GM-CSF, IFN-ɣ, IL-10, IL-2, IL-4, IL-6, TNF, IL-8 and IL-1β is shown. The inferno colour scale indicates the abundance of each cytokine in each subject, from black (zero, not observed) to very light yellow (highly abundant).

**Figure 5 ijms-21-08218-f005:**
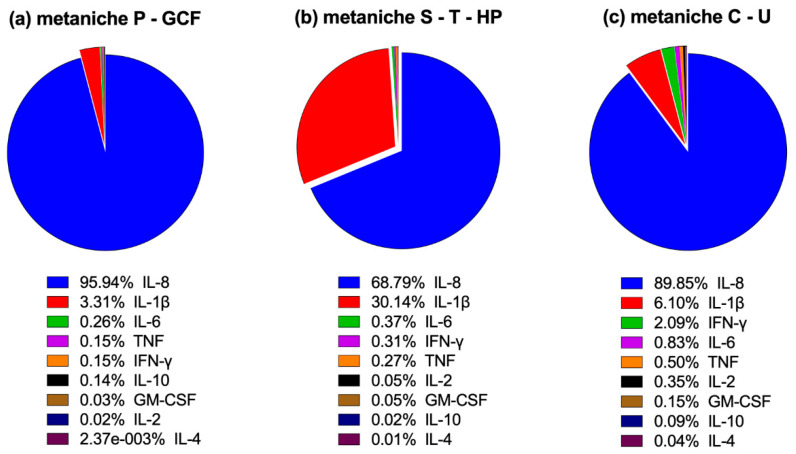
Relative quantities of cytokines in the defined microbial metaniches P-GCF, S-T-HP and C-U are given as a percentage of the total cytokine level measured. (**a**) metaniche P-GCF, (**b**) metaniche S-T-HP and (**c**) metaniche C-U.

**Figure 6 ijms-21-08218-f006:**
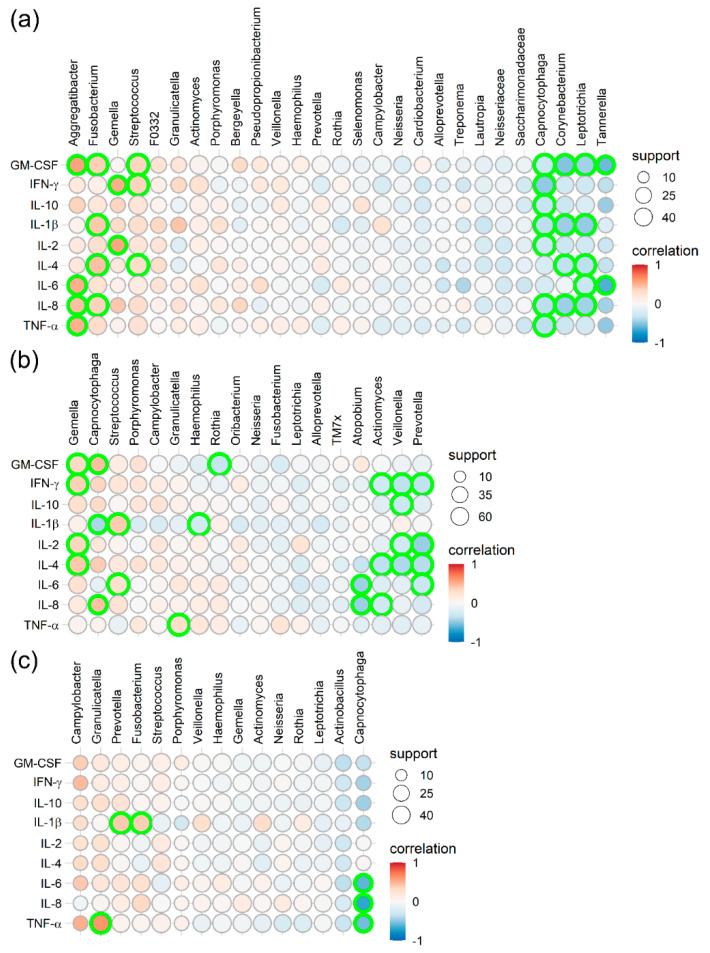
Pearson correlation plot between the cytokine concentrations and microbial composition (genus level) are given for the defined metaniches: (**a**) “plaque and gingival crevicular fluid” (P-GCF), (**b**) “saliva, tongue and hard palate” (S-T-HP) and (**c**) “cheek-sublingual” (C-U). The cytokines GM-CSF, IFN-γ, IL-10, IL-1ß, IL-2, -4, IL-6, IL-8, and TNF are given vertically on the left side, while genera are indicated horizontally on top of each figure. A colour key was used for the strength of correlation, ranging from highly positive (+1, red) to no correlation (0, white) to highly negative (−1, blue). The number of samples on which the results are based is depicted by the size of the circles, with a minimum of 10 and a maximum of 40 or 60 samples per metaniche. Shown are genera with a significant Pearson correlation (*p* < 0.05). Samples with significant correlation after Benjamin–Hochberg adjustment for sample size (adj. *p* < 0.05) were highlighted with bright green circles.

**Table 1 ijms-21-08218-t001:** List of inclusion and exclusion criteria for subject enrolment in the clinical study.

Inclusion	Exclusion	
young adults male and female age: 20–35 years excellent oral health dentistry students, dentists, orthodontists or scientific assistants of the university hospital of Erlangen signed declaration of consent by the patient	systemic or metabolic disease that are related to gingivitis (e.g., diabetes) or could possibly influence the oral microbiota gingivitis or periodontitis (PSI ≥ 3) missing teeth due to extraction because of carious lesions more than 6 missing teeth (including third molar extractions or hypodontia/congenitally missing teeth) active carious lesions, peri-/apical lesions, fistulas, oral abscesses pregnancy, regular smoking	presence of orthodontic appliances, except fixed retainers to the incisors obesity: body mass index (BMI) > 30 kg/m² eating disorder or underweight: BMI < 18.5 kg/m² intake of antibiotics or dietary supplementation in the last 6 months disconfirmation of the declaration of consent by the patient

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
