# Peer review of "Defining Metaniches in the Oral Cavity According to Their Microbial Composition and Cytokine Profile"

_ijms, 2020, doi:10.3390/ijms21218218_

Round 1

Reviewer 1 Report

Although the methods and conclusions from this study are sound, the manner by which the data are presented is very lacking. This is an incredibly confusing and overly figure heavy document. I suggest revisions (I would say many minor revisions) including the restructuring and removal of some figures and content from this publication, much of which can go into the supplemental material. 

Introduction- the authors go into depth regarding microbiome analysis and oddly mention databases. Anyone in the microbiome field knows these databases exist, I'm not sure why they're being discussed or what the '775 taxa .. detected' refers to (which database- silva or HOMD?). Despite their detail in the microbiome field, they spend zero time discussing cytokines, which is an important piece of this work. Why overly discuss microbiome but make no mention of what biomarkers are important and why? At least another paragraph is needed here to tell the reader why this is worth examining. 

Results- Table 1 is very badly put together. Why are the data being presented like this and not in a proper bar graph or sensical chart form? The numbers are scattered all throughout the chart instead of in one single column for the reader to easily digest. This MUST be fixed. Additionally, these charts are followed by long chunks of listed text that have no business being in the main text. These should be either put into another chart or delegated to the supplemental area. The authors do this once again after figure 2. 

Figure 1 and Figure 3 are perfect examples of presenting TOO much data. This was very confusing as a reader, being bombarded with all this information. In my opinion, only the content critical to the story should be displayed in the main text. The authors also discuss metaniches in Figure 2 and 3 but then revert back to normal niches for figure 4? Additionally, is it really critical to have both figures 5 and 6 present? Perhaps a heatmap for the whole cohort would work here. 

Although I do this this work is important, I found this work incredibly difficult to read and bogged down by too many inconsequential figures and unnecessary variables. This seems to be three separate studies jammed into one. Due to this, it seems like the thesis statement of the project is muddled. 

Author Response

Response to Reviewer 1 Comments

Point 1

Although the methods and conclusions from this study are sound, the manner by which the data are presented is very lacking. This is an incredibly confusing and overly figure heavy document. I suggest revisions (I would say many minor revisions) including the restructuring and removal of some figures and content from this publication, much of which can go into the supplemental material.

We thank the reviewer for the suggestions and followed the advice. To improve comprehensibility, we restructured and levelled down the results part, created new figures and put some figures/tables into the supplemental material (Figure S1, S2 and Table S1) as explained in detail below.

Point 2

Introduction- the authors go into depth regarding microbiome analysis and oddly mention databases. Anyone in the microbiome field knows these databases exist, I'm not sure why they're being discussed or what the '775 taxa .. detected' refers to (which database- silva or HOMD?). Despite their detail in the microbiome field, they spend zero time discussing cytokines, which is an important piece of this work. Why overly discuss microbiome but make no mention of what biomarkers are important and why? At least another paragraph is needed here to tell the reader why this is worth examining.

We understand the remark that databases should not be explained in detail. Therefore, we deleted the sentence explaining database HOMB and SILVA (l. 61). We apologize for the unclear explanation concerning of “775 taxa”: We refer to the up-to-date numbers of SILVA database.

Moreover, we thank the reviewer for the suggestion to add more studies about biomarkers.

Therefore, we added a longer paragraph (ll. 75-87) about the role of cytokines in periodontitis as well as in oral diseases. Taken together, 9 references (20-29) about the role of cytokines are given including 3 studies focusing on the role of salivary cytokine biomarkers in periodontitis and oral squamous cell carcinoma (smoking as a predisposing factor).

Point 3

Results- Table 1 is very badly put together. Why are the data being presented like this and not in a proper bar graph or sensical chart form? The numbers are scattered all throughout the chart instead of in one single column for the reader to easily digest. This MUST be fixed. Additionally, these charts are followed by long chunks of listed text that have no business being in the main text. These should be either put into another chart or delegated to the supplemental area. The authors do this once again after figure 2.

We apologize for the presentation of table 1. We created a new figure (Fig. S1) in cake chart form for better readability. As Figure S1 is not mandatory for the understanding and definition of metaniches, we put the new figure in the supplemental area.

As for the “long chunks of listed text in the main text”, we shortened the results part:

1)       As Figure 1 contains detailed data about the relative distribution of genera in oral niches, we deleted the listed relative percentual values from the main text (ll. 142-151).

2)       As Figure 3 presents the most distinct OTUs for the defined metaniche P-GCF, S-T-HP and C-U, we deleted the listed relative percentual values from the main text (ll. 344--617), which follows Figure 2.

Point 4

Figure 1 and Figure 3 are perfect examples of presenting TOO much data. This was very confusing as a reader, being bombarded with all this information. In my opinion, only the content critical to the story should be displayed in the main text. The authors also discuss metaniches in Figure 2 and 3 but then revert back to normal niches for figure 4? Additionally, is it really critical to have both figures 5 and 6 present? Perhaps a heatmap for the whole cohort would work here.

We understand that Figure 1 and Figure 3 contain much data. However, we need to mention that in total 280 samples were analysed. Therefore, the dendrogram, the genus frequencies plot and Shannon index representing the genera detected in different oral niches of all sampled sites appears very long (Figure 1). To reduce data and allow better readability, we deleted the listed relative percentual values from the main text (ll. 142-151). As the definition of metaniches is based on the observed clustering (ll-132-137) and on the significant differences regarding the beta diversity between P-GCF, S-T-HP and C-U (ll.151-13), we believe that the presentation of Figure 1 is indeed necessary content and critical to the story. Moreover, the presentation of dendrogramms is a standard presentation of the microbial complexity (e.g. Ravel J, Gajer P, Abdo Z, et al. Vaginal microbiome of reproductive-age women. Proc Natl Acad Sci U S A. 2011;108 Suppl 1(Suppl 1):4680-4687. doi:10.1073/pnas.1002611107).

Figure 3 presents the most distinct OTUs (24 in total) for the defined metaniche P-GCF, S-T-HP and C-U, which we believe is not only important for the content of the story but also very interesting for future studies. However, to level down data in the results part, we deleted the listed relative percentual values from the main text (ll. 344--617).

We understand that it might be irritating to revert back to normal niches in Figure 4. But investigation of cytokine concentrations in all oral niches was necessary to analyze whether the detected microbial metaniches can also be found in the cytokine pattern. We support the idea of putting some figures in the supplemental material. Therefore, we shifted Figure 4 from the main text to the supplemental material (Figure S2).

Moreover, we want to thank the reviewer for the suggestion to create a heat map for the whole cohort. Therefore, we added IL-8 and IL-1β to the heat maps (Figure 4, formerly Figure 6). Moreover, we shortened the results part for better readability.

To point out similarities in the cytokine profile similar to microbial metaniches, we followed the reviewer’s advice and replaced Figure 5 (showing relative quantities of cytokines for each singular niche) with a small Figure (Figure 5) presenting relative quantities of cytokines in the defined microbial metaniches P-GCF, S-T-HP and C-U.

Point 5

Although I do this this work is important, I found this work incredibly difficult to read and bogged down by too many inconsequential figures and unnecessary variables. This seems to be three separate studies jammed into one. Due to this, it seems like the thesis statement of the project is muddled.

We thank the reviewer for the criticism and apologize for the presentation of our results. We followed the advice to highlight our statements – the definition of metaniches. Therefore, some Figures were put in the supplemental material and some were recreated. Moreover, the main text was revised.

We like to point out that the investigation of both the microbial composition and the cytokine profile as well as the correlation of both parts is important in analyzing the complexity of oral niches. We are convinced that the direct comparison between microbiomes and cytokine composition in different oral niches is an important step in understanding development of complex inflammatory diseases like periodontitis.

We are sorry that our presentation of results was too complicated. We believe that the changes suggested by the reviewer helped to streamline the manuscript and rigorously improved readability.

Reviewer 2 Report

Defining metaniches in the oral cavity according to 3 their microbial composition and cytokine profile

Seidel et al., submitted to International Journal of Molecular Sciences

This manuscript describes the outcomes of a study examining and relating the microbiomes and cytokine contents of various oral niches of healthy volunteer young adults.

Microbiome content was assessed using 16S rDNA sequencing and cytokine profiles were determined using ELISA. Statistical comparison were used to determine niches with common microbiome and cytokine characteristics.

Overall the study was well conducted and provides a dataset for this cohort (n=20) upon which future work can be developed.

The Abstract is clear and describes the aims and outcome of the study.

The Introduction provides sufficient information to introduce the study. More than one reference to studies describing investigation of salivary cytokines as biomarkers should be made. Currently only reference [16] is given. 

Results. 2.1 Defining of the niche abbreviations should be made here. Furthermore a brief description of the sampling process would be helpful, eg “Sample were obtained by swap of the hard palate (HP), cheek (C), sublingual surface (U)….. etc.  

Methods: Statistics - please remove reference to P-values as “very significant”, “extremely significant”, “highly significant”. This is nonsense. The P-value indicates the likelihood that an observation occurs due to chance, with the accepted “significance” value arbitrarily defined for each study. It does not convey anything about ranking of importance.

Discussion. The discussion related the data to observations of others reported in the literature. Discussion of the results and implications with respect to periodontal disease was made. A brief discussion of the emerging data relating oral cytokine levels and the microbiome to other oral conditions, such carcinoma, would have added some interest as a future direction if the study approach.

Some small formatting/typographical issues:

All genus (in the singular) and species names should be in italics.  

Line 182 and 183: no SEM is given for the S-T-HP group OTU.

Line 245, GCF cytokine content is reported as “13,28  ±2,87 pg/mL”. The last digit is missing from each value.

Reference formatting is inconsistent. For examples, article title should be sentence case and not have non-proper noun words in capitals. Sometimes journal names were abbreviated but other time not and use of capitals was inconsistent; etc.

Author Response

Response to Reviewer 2 Comments

Point 1

Defining metaniches in the oral cavity according to 3 their microbial composition and cytokine profile. Seidel et al., submitted to International Journal of Molecular Sciences. This manuscript describes the outcomes of a study examining and relating the microbiomes and cytokine contents of various oral niches of healthy volunteer young adults. Microbiome content was assessed using 16S rDNA sequencing and cytokine profiles were determined using ELISA. Statistical comparison were used to determine niches with common microbiome and cytokine characteristics. Overall the study was well conducted and provides a dataset for this cohort (n=20) upon which future work can be developed. The Abstract is clear and describes the aims and outcome of the study.

We thank the reviewer for the appreciation and positive review.

Point 2

The Introduction provides sufficient information to introduce the study. More than one reference to studies describing investigation of salivary cytokines as biomarkers should be made. Currently only reference [16] is given.

We thank the reviewer for thesuggestion to add more studies about salivary cytokines. We followed the advice and added a longer paragraph (ll. 75-87) about the role of cytokines in periodontitis as well as in oral diseases. Taken together, 9 references (20-29) about the role of cytokines are given including 3 studies focusing on the role of salivary cytokine biomarkers in periodontitis and oral squamous cell carcinoma (smoking as a predisposing factor).

Point 3:

Results. 2.1 Defining of the niche abbreviations should be made here. Furthermore a brief description of the sampling process would be helpful, eg “Sample were obtained by swap of the hard palate (HP), cheek (C), sublingual surface (U)….. etc.

We thank the reviewer for the suggestion and followed the advice. To allow a better understanding of the results part, we put in a short paragraph (ll- 104-110) at the beginning of the results including a brief description of the sampling process as well as the definition and abbreviation of the considered oral niches. A detailed description of the sampling process is given in the section “material & methods” (4.s. sample collection).

Point 4

Methods: Statistics - please remove reference to P-values as “very significant”, “extremely significant”, “highly significant”. This is nonsense. The P-value indicates the likelihood that an observation occurs due to chance, with the accepted “significance” value arbitrarily defined for each study. It does not convey anything about ranking of importance.

We thank the reviewer for the advice and removed the references to the P-values and added the following definition of the p-value: “Differences were considered significant with P-values < 0.05” (l 1329).

Point 5

Discussion. The discussion related the data to observations of others reported in the literature. Discussion of the results and implications with respect to periodontal disease was made. A brief discussion of the emerging data relating oral cytokine levels and the microbiome to other oral conditions, such carcinoma, would have added some interest as a future direction if the study approach.

We thank the reviewer for the remark about studies regarding the role of oral cytokine levels and the composition of the oral microbiota in oral cancer. Accordingly, we added a paragraph (ll 1170-1177) at the beginning of the section “correlation between microbiota composition and cytokine expression”. Moreover, 3 references (27,48,28) were added to highlight the importance of future research considering both oral microbiome as well as oral cytokine levels to gain more knowledge not only in the field of periodontitis but also regarding oral diseases like oral squamous cell carcinoma.

Point 6

Some small formatting/typographical issues:All genus (in the singular) and species names should be in italics. Line 182 and 183: no SEM is given for the S-T-HP group OTU. Line 245, GCF cytokine content is reported as “13,28 ±2,87 pg/mL”. The last digit is missing from each value.

We apologize for small formatting errors and thank the reviewer for this notation. All genus and species, which were not in italics so far, were formatted (e.g. ll 341-596). Line 182 and 183 were removed as reviewer 1 suggested to shorten and modify the results part. The GCF cytokine content is now given with 2 digits: 13284.78 ± 2869.64 pg/mL (l 634). All genera and species were formatted into italics.

Point 7

Reference formatting is inconsistent. For examples, article title should be sentence case and not have non-proper noun words in capitals. Sometimes journal names were abbreviated but other time not and use of capitals was inconsistent; etc.

We thank the reviewer for the remark. Article titles were formatted into sentence case and journal names were abbreviated.

Overall,we thank the reviewer for the constructive criticism, which unequivocally improved our manuscript!